# LightAgent: Lightweight and Cost-Efficient Mobile Agents

## Abstract

$\tau$ With the advancement of multimodal large language models (MLLMs), building GUI agent systems has become an increasingly promising direction—especially for mobile platforms, given their rich app ecosystems and intuitive touch interactions. Yet mobile GUI agents face a critical dilemma: truly on-device models (4B or smaller) lack sufficient performance, while capable models (starting from 7B) are either too large for mobile deployment or prohibitively costly (*e.g.*, cloud-only closed-source MLLMs). To resolve this, we propose LightAgent, a mobile GUI agent system that leverages device-cloud collaboration to tap the cost-efficiency of on-device models and the high capability of cloud models, while avoiding their drawbacks. Specifically, LightAgent enhances Qwen2.5-VL-3B via two-stage SFT→GRPO training on synthetic GUI data for strong decision-making, integrates an efficient long-reasoning mechanism to utilize historical interactions under tight resources, and defaults to on-device execution—only escalating challenging subtasks to the cloud via real-time complexity assessment. Experiments on the online AndroidLab benchmark and diverse apps show LightAgent matches or nears larger models, with a significant reduction in cloud costs. We have made our LightAgent available anonymously at: https://anonymous.4open.science/r/LightAgent-E2D5/.

## 1 Introduction

The growing capability of multimodal large language models (MLLMs) enables AI agents to perceive and act within visual environments, particularly through Graphical User Interfaces (GUIs) (Wu et al., 2024b; Qi et al., 2024). Mobile platforms offer a promising domain for this technology for two reasons: first, their vast app ecosystems provide a realistic and diverse testbed, and second, their touchscreen interactions are limited to an intuitive set of primitives, resulting in a compact action space. Despite these advantages, mobile platforms introduce distinct challenges, chiefly severe computational and memory limitations. In light of these factors, our goal is to develop an effective mobile GUI agent, viewing it as a practical milestone toward general-purpose AI.

Current research falls broadly into two groups. The first involves targeted training of open-source MLLMs specifically for GUI-related tasks (Qin et al., 2025; Dai et al., 2025; Liu et al., 2024). These models are relatively compact in size and have achieved notable progress; for instance, UI-Tars-7B (Qin et al., 2025) outperforms the larger Qwen2.5-VL-32B (Bai et al., 2025) on mobile GUI tasks. The second approach leverages general-purpose closed-source MLLMs by constructing multi-agent systems and designing well-structured execution pipelines. Thanks to the powerful multimodal comprehension capabilities of state-of-the-art (SOTA) closed-source MLLMs—such as GPT-5 (OpenAI, 2025), Claude-Sonnet-4 (Anthropic, 2025), and Gemini-2.5-Pro (Comanici et al., 2025)—their performance on mobile GUI tasks can even surpass that of models specifically trained for such tasks. However, there is no free lunch. Although the performance of models like UI-Tars-7B is impressive given their 7B scale, MLLMs of this size still impose a prohibitive computational burden on contemporary smartphones (Laskaridis et al., 2024). A more practical 2B–3B scale, by contrast, typically yields MLLMs with limited capabilities (Lin et al., 2025). On the other hand, multi-agent systems based on advanced closed-source MLLMs are plagued by high costs. SOTA closed-source MLLMs are often expensive, and spending several to dozens of dollars to complete a single mobile task is financially impractical. The detailed discussion of related work is in Appendix A.2.

In response to the aforementioned challenges, two immediate questions arise: **Question 1:** For task-specific GUI models, can their size be further reduced to become truly on-device models capable of running on smartphones, while maintaining acceptable performance levels on GUI tasks? **Question 2:** For proprietary general-purpose large models, which are indispensable as cloud-based models due to their powerful capabilities, can their usage costs be further reduced?

To answer to the aforementioned questions, we propose a lightweight device-cloud collaborative mobile GUI agent framework—LightAgent. Specifically, for **Question 1**, based on a lightweight open-source MLLM (*i.e.*, Qwen2.5-VL-3B), we employ a synthetic data generation pipeline and conduct two-stage training comprising supervised fine-tuning (SFT) and group relative policy optimization (GRPO), resulting in a compact yet powerful on-device GUI agent that achieves performance comparable to larger-scale models. Concurrently, we design an efficient long-reasoning GUI agent paradigm, which is capable of effectively processing historical operation information and endowing the agent with reasoning capabilities to further enhance its performance.

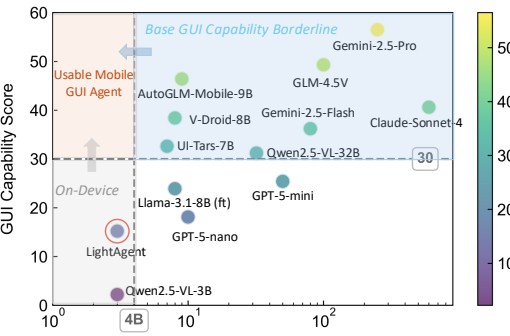

Figure 1: Model GUI Capability vs. Cost. Methods in the on-device (gray) region lack usable GUI capability, while basic-GUI-capability (blue) ones are too large for on-device deployment or too costly. Currently, there are no suitable approaches in the truly usable mobile GUI agent (orange) region. A promising research direction is combining gray and blue region methods, leveraging their complementary strengths to bridge the gap for practical on-device GUI agents.

For **Question 2**, we observe that advanced general-purpose large models exhibit performance overkill on many simple GUI tasks—tasks that even small on-device models can handle competently. Moreover, for tasks that small models cannot fully complete, they often fall short only by a narrow margin. To this end, we devise a device-cloud collaboration paradigm that leverages the cost-efficiency of on-device models while compensating for their performance limitations through the powerful capabilities of cloud-based models, thereby identifying a sweet spot between performance and overhead. Through the task complexity assessment and a dynamic orchestration policy, our framework enables real-time monitoring of task execution progress and dynamic switching between on-device and cloud models as needed. Our main contributions can be summarized as follows:

- **Light-weight reasoning GUI agent.** We develop a lightweight on-device GUI agent by applying a two-stage training methodology to a compact MLLM, equipping it with efficient long-reasoning capabilities to contextualize historical interactions for effective decision-making. The resulting model achieves competitive performance with a minimal computational footprint, suitable for smartphone deployment.

- **Device-cloud collaborative agent system.** We propose a device-cloud collaborative agent system that dynamically orchestrates tasks between on-device and cloud models via real-time complexity assessment. This mechanism enables seamless switching to the cloud only when necessary, compensating for on-device limitations, achieving an optimal balance between high performance and significantly reduced operational costs.

- **Comprehensive Evaluation**. We evaluate LightAgent through online experiments on the Android-based AndroidLab benchmark, complemented by dozens of custom tasks across popular apps to assess real-world performance. Furthermore, we conduct extensive ablation studies and investigate the impact of different fine-tuning strategies on the lightweight MLLM's capabilities.

## 2 METHODOLOGY

To effectively address mobile GUI agent tasks, we propose the LightAgent framework, which has three key modules. Section 2.1 covers strategies to mitigate compact MLLMs' challenges in GUI tasks—limited capacity and constrained context length. Section 2.2 details a device-cloud collaborative agent system that dynamically schedules on-device and cloud models by task difficulty, balancing cloud resource consumption with GUI task completion rate. Section 2.3 outlines ways to fully leverage limited training data to maximize small MLLMs' performance gains on GUI tasks.

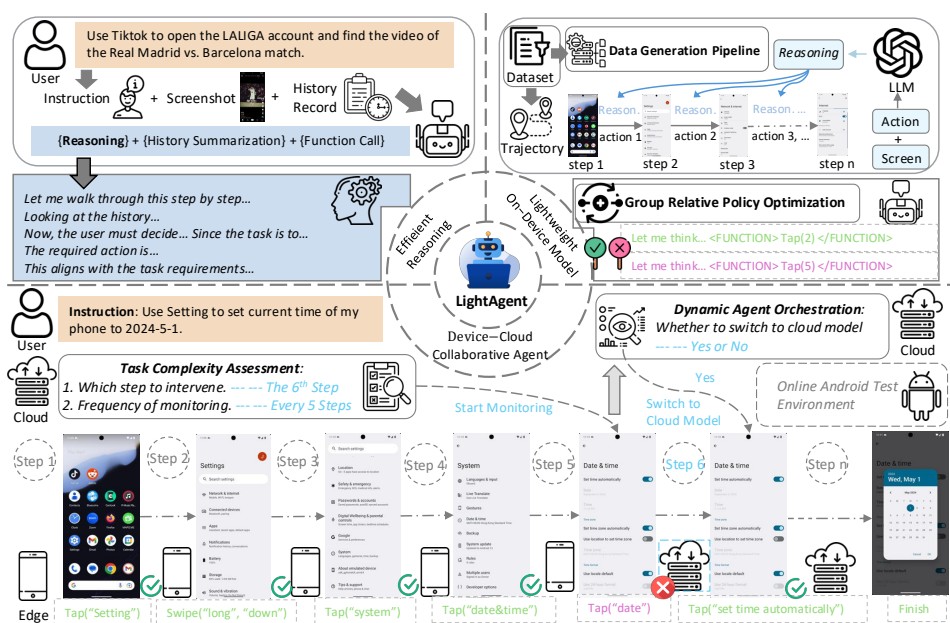

Figure 2: Overall framework of the proposed LightAgent.

## 2.1 EFFICIENT REASONING GUI AGENT

The on-device GUI agent faces two key challenges. First, mobile devices' limited computing power necessitates small-sized MLLMs (*e.g.*, Qwen2.5-VL-3B (Bai et al., 2025)), which currently lack sufficient performance for mobile GUI tasks. To tackle this, LightAgent enhances the GUI agent with extended chain-of-thought (CoT) reasoning (Wei et al., 2022) during testing, using test-time scaling laws (Snell et al., 2024) to boost its capability. Second, limited on-device resources impede long-context handling: high-resolution GUI images take up much available context length, and managing the agent's execution history is challenging. To alleviate this, LightAgent uses an efficient text-based summarization scheme—compressing each step's state into compact textual representations—to support the agent's long historical context. The output format template is presented in Figure 3, and the detailed instruction template is provided in Appendix A.4.1.

### 2.1.1 LONG-HORIZON REASONING ENHANCEMENT

Mobile GUI tasks are often difficult and complex, and humans also use step-by-step reasoning for such operations. Motivated by the success of CoT reasoning and test-time scaling laws, it is natural to apply similar long-form reasoning enhancements to GUI agents.

Specifically, the GUI agent's reasoning—encompassing all factors for task completion—follows a multi-step process: First, it analyzes actionable elements on the current interface and, using historical data, assesses if prior actions met their goals. Next, it evaluates progress toward the user's task goal and identifies necessary follow-up actions. Lastly, it selects from available functions and their parameters to generate the final output. Notably, if historical data shows prior operations failed to yield expected results, the model proactively reflects and adjusts its approach—avoiding repeated errors and potential loops.

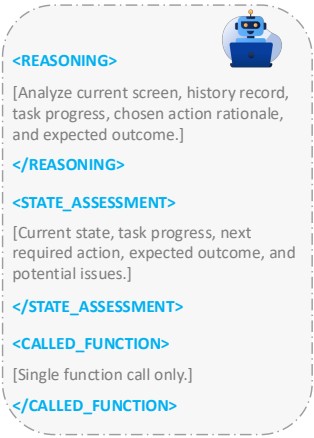

Figure 3: Output Template.

### 2.1.2 EFFICIENT MEMORY MANAGEMENT

On-device GUI agents also face the challenge of efficiently managing historical contextual information. For MLLMs in particular, high-resolution images (which preserve valuable details) require a

large number of tokens—making raw screenshot storage in history impractical. As a result, systems can only retain a limited number of recent images, leading to the loss of long-term historical data.

To address this, LightAgent uses a textual summarization approach: at each step, it distills all information relevant to future actions. As shown in the `<STATE_ASSESSMENT>` field of Figure 3, these summaries include the current interface state, task progress, the agent's inferred next action, expected post-action outcome, and potential issues. Much of this content comes from condensing the `<REASONING>` section—critical for effective stepwise reasoning, especially reflective error correction. Crucially, textual summaries use far fewer tokens than images, enabling long-term history retention (*e.g.*, 10–20 steps) as contextual input, even in resource-constrained mobile environments.

### 2.1.3 OVERALL PROCESS

Formally, the process is a sequential decision-making framework. At each time step $t \in \mathbb{N}_0$ (with $t = 0$ denoting the initial step) the agent receives a task instruction $\tau \in \mathcal{T}$ and observes a screen screenshot $s_t \in \mathcal{S}$. The history $h_t$ belongs to $\mathcal{H} = \bigcup_{k=0}^{\infty} \mathcal{A}^k$, where $\mathcal{A}^0 = \{\epsilon\}$ and $\epsilon$ denotes the empty sequence. The history $h_t$ is the sequence of previous state assessments $a_k \in \mathcal{A}$ for $k < t$; each assessment $a_t \in \mathcal{A}$ is a structured summary of the interface state, task progress, the next action, the expected outcome, and potential issues. When $t = 0$ we have $h_0 = \epsilon$ (no historical data).

The reasoning function R maps the current history, the current observation, and the task instruction to a new assessment and a function to execute:

$$R : \mathcal{H} \times \mathcal{S} \times \mathcal{T} \to \mathcal{A} \times \mathcal{F}, \tag{1}$$

$$(a_t, f_t) = R(h_t, s_t, \tau), \qquad a_t \in \mathcal{A}, \ f_t \in \mathcal{F}, \tag{2}$$

where $\mathcal{F}$ is the space of executable functions. The history is then updated by concatenation:

$$h_{t+1} = h_t \circ a_t, \tag{3}$$

with $\circ$ denoting sequence concatenation and $h_0 = \epsilon$. The process repeats for $t = 0, 1, 2, \ldots$ until the task is completed or a predefined termination condition is met. We report the action space of LightAgent in Appendix A.3.

## 2.2 DEVICE-CLOUD COLLABORATIVE AGENT SYSTEM

Despite recent advances in small-scale MLLMs, their performance remains insufficient for handling complex GUI-based tasks. As illustrated in Figure 4, even the GUI model UI-Tars-7B—fine-tuned on a substantial amount of GUI task data—exhibits significantly poorer performance compared to larger cloud-based models. Furthermore, the performance of more deployment-viable 3B-parameter models (*e.g.*, Qwen2.5-VL-3B) falls well below a practically usable threshold.

This limitation necessitates the incorporation of more powerful cloud-based LLMs (such as Gemini-2.5-Pro, GPT-5, or Claude-Sonnet-4) to achieve satisfactory task completion and user experience. However, frequent invocation of cloud models leads to high operational costs. A careful analysis of on-device model failures reveals that many tasks fail only at the final step. Motivated by this observation, we propose a collaborative device-cloud agent system that dynamically orchestrates between local and cloud agents based on real-time task progress. This approach significantly reduces cloud invocations and associated costs while maintaining high task success rates.

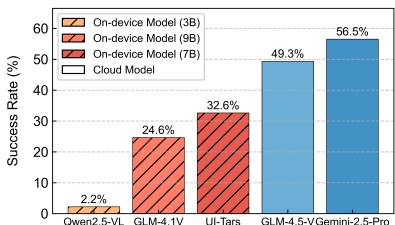

Figure 4: GUI Performance on AndroidLab: On-Device Models vs. Cloud Models.

The system functions via an integrated workflow with two core components: a **task complexity assessment mechanism** (to decide when and how often to monitor agent performance) and a **dynamic orchestration policy** (to trigger agent switching when needed). The entire process is summarized in Algorithm 1, which merges these two components into a unified adaptive framework.

### 2.2.1 COLLABORATIVE CONTROL FRAMEWORK

**Task Complexity Assessment**. Before task execution begins, LightAgent estimates the difficulty of the task using aggregated historical performance data of the on-device model. An assessment function

$f_{\text{assess}}$ analyzes the task description and context to determine two key parameters: the step $\gamma$ at which monitoring should begin, and the monitoring interval $\omega$. This preemptive configuration allows the on-device agent to fully utilize its capability without premature—and costly—cloud intervention.

**Dynamic Orchestration Policy**. During task execution, the system monitors the agent's behavior once the step counter reaches $\gamma$ and at every $\omega$ steps thereafter. A switching function $\mathcal{F}_{\text{switch}}$ evaluates the current GUI state and execution history—including previous actions, state transitions, and task progress—against three criteria: (1) presence of repetitive action patterns, (2) deviation from the expected task trajectory, or (3) inadequate action quality. If any criterion is met, the system switches the current agent from the on-device model $\mathcal{M}_{\text{device}}$ to the cloud model $\mathcal{M}_{\text{cloud}}$, and no further monitoring is performed. This mechanism minimizes unnecessary cloud calls while ensuring reliability through conditional fallback to a more capable cloud model.

The instruction templates for these two components are reported in Appendices A.4.2 and A.4.3.

### 2.2.2 INTEGRATED EXECUTION FLOW

The overall workflow of the device-cloud collaborative agent system is detailed in Algorithm 1. The algorithm begins by invoking $\mathcal{F}_{\text{assess}}$ to determine $\gamma$ and $\omega$. The execution loop uses the current agent $\mathcal{M}_{\text{current}}$ (initialized to $\mathcal{M}_{\text{device}}$) to perform the function $f$ and update the assessment $a$ and the history $h$. If the step condition is satisfied and switching has not yet occurred, $\mathcal{F}_{\text{switch}}(\cdot)$ is evaluated. Upon switching, the cloud agent takes over and continues until task completion. This approach ensures cost-efficient execution while maintaining robust performance.

---

**Algorithm 1:** Adaptive Device-Cloud Agent Switching Algorithm

---

**Input:** Task $\tau$, On-device agent $\mathcal{M}_{\text{device}}$, Cloud agent $\mathcal{M}_{\text{cloud}}$
**Output:** State sequence $\{s_n\}$, History $\{h_n\}$

1   $(\gamma, \omega) \leftarrow \mathcal{F}_{\text{assess}}(\tau, \mathcal{M}_{\text{device}})$;      // $\gamma$: `monitoring start step`, $\omega$: `monitoring`
      `frequency`
2   $t \leftarrow 0$, $\mathcal{M}_{\text{current}} \leftarrow \mathcal{M}_{\text{device}}$, $C_{\text{switched}} \leftarrow$ false, $C_{\text{completed}} \leftarrow$ false, $C_{\text{terminated}} \leftarrow$ false;
3   **while** $\neg C_{completed} \wedge \neg C_{terminated}$ **do**
4     **if** $\neg C_{switched} \wedge (t \geq \gamma) \wedge ((t - \gamma) \mod \omega == 0)$ **then**
5       **if** $\mathcal{F}_{switch}(a_t, f_t, h_t) == \textit{True}$ **then**
6         $\mathcal{M}_{\text{current}} \leftarrow \mathcal{M}_{\text{cloud}}$;
7         $C_{\text{switched}} \leftarrow$ true;
8       **end**
9     **end**
10    $a_t, f_t \leftarrow \mathcal{M}_{\text{current}}(h_t, s_t, \tau)$;                                       // Eq. 2
11    $h_{t+1} \leftarrow h_t \circ a_t$;                                                 // Eq. 3
12    $t \leftarrow t + 1$;
13    update($C_{\text{completed}}, C_{\text{terminated}}$);    // `Update` $C_{\text{completed}}$ `and` $C_{\text{terminated}}$ `via environment`
14 **end**

---

### 2.3 LIGHTWEIGHT MLLM TUNING FOR ON-DEVICE AGENTS

Unlike existing GUI agent methods, we use a smaller MLLM (*i.e.*, Qwen2.5-VL-3B) to mitigate mobile resource constraints and boost practicality. However, fine-tuning such a compact MLLM for usable GUI agent performance poses notable challenges. Small MLLMs naturally have limited capabilities; to enhance their GUI manipulation ability, we leverage the test-time scaling law—via long-chain reasoning during inference—to improve performance, as detailed in Section 2.1.1.

A key challenge in GUI agent training is the scarcity of high-quality data, which depends heavily on expensive manual annotation. To tackle this, we design an **automated synthetic data pipeline**: it optimally uses limited human-annotated examples to generate augmented instances with explicit reasoning chains. Using this generated data, we propose a **two-stage fine-tuning paradigm** to elicit long-chain reasoning in small MLLMs, allowing them to analyze, reflect, and ultimately generate high-quality responses.

### 2.3.1 SYNTHETIC DATA GENERATION PIPELINE

Human-annotated GUI trajectory datasets usually include only task instructions, screen snapshots, and ground-truth actions. Lightweight MLLMs struggle to gain long-chain reasoning and reflective capabilities from this limited supervision. Thus, high-quality data with explicit reasoning chains are essential to activate their reasoning capacity. Based on this, we design a data-generation pipeline.

Specifically, we first use an advanced MLLM (*e.g.*, Gemini-2.5-Pro) to generate chain-of-thought reasoning using the task instruction, target function, and historical interaction context. A powerful LLM (*e.g.*, Qwen3-32B) then uses this MLLM-generated reasoning, along with the original task instruction, to synthesize the needed training instances, as specified in Figure 3. The instruction template for reasoning data generation is provided in Appendix A.4.4.

### 2.3.2 TWO-STAGE TRAINING PROTOCOL

Model training has two stages: supervised fine-tuning (SFT) followed by group relative policy optimization (GRPO) (Shao et al., 2024). In the first stage, chain-of-thought annotations in synthetic training data impart basic reasoning skills and foundational GUI task competence to the small MLLM. This supervised grounding generates meaningful intermediate behaviors, preventing the subsequent reinforcement learning stage from facing overly sparse or uninformative rewards. The second stage is a reinforcement-style policy optimization phase: well-designed reward functions here directly boost the correctness of the model's output actions and align its behavior with GUI task completion goals.

**Reward Design.** In the GRPO algorithm, the total reward $\mathcal{R}_{\text{total}}$ combines accuracy and format components as defined in the following equation:

$$\mathcal{R}_{\text{total}} = \underbrace{\mathcal{R}_{\text{acc}} \cdot \begin{cases} 1, & \text{if } f_{\text{pred}} = f_{\text{gt}} \text{ (operations)} \\ 1, & \text{if } \text{sim}(a_{\text{pred}}, a_{\text{gt}}) \geq \lambda \text{ (queries)} \\ 0, & \text{otherwise} \end{cases}}_{\text{Accuracy Reward } \mathcal{R}_{\text{accuracy}}} + \underbrace{\mathcal{R}_{\text{fmt}} \cdot \frac{k}{3} \cdot \psi^c}_{\text{Format Reward } \mathcal{R}_{\text{format}}} \tag{4}$$

$R_{acc}$ and $R_{fmt}$ denote the rewards for answer accuracy and formatting correctness, respectively, during reinforcement learning. By default, both values are set to 1.

(i) **Accuracy Reward.** $\mathcal{R}_{\text{accuracy}}$ is task-dependent: for *operation* tasks like `"Tap(index)"`, it requires strict matching between predicted output $f_{\text{pred}}$ and ground truth $f_{\text{gt}}$, while for *query* tasks like `"Finish(answer)"`, it employs embedding-based similarity between predicted answer $a_{\text{pred}}$ and ground truth $a_{\text{gt}}$, granting reward only when $\text{sim}(a_{\text{pred}}, a_{\text{gt}}) \geq \lambda$, with the reward being 0 when these conditions are not satisfied.

(ii) **Format Reward.** $\mathcal{R}_{\text{format}}$ provides a base reward of $\mathcal{R}_{\text{fmt}} \cdot \frac{k}{3}$ based on adherence to Figure 3's three-block structure, where $k$ measures the degree of conformity (full reward requires complete adherence with $k = 3$), and applies a multiplicative penalty $\psi^c$ with coefficient $\psi < 1$ based on the amount of content $c$ outside the template, a mechanism specifically designed to mitigate irrelevant generation common in small-scale MLLM training.

**GRPO Training.** GRPO eliminates the need for additional value function approximation, as seen in PPO (Schulman et al., 2017), and instead utilizes the average reward from multiple sampled outputs generated in response to the same question as its baseline. Specifically, for each question $q \sim \mathcal{Q}$, a group of outputs $\{o_1, o_2, \ldots, o_G\}$ is sampled from the old policy $\pi_{\text{old}}$. The model is optimized by maximizing the following objective:

$$J(\theta) = \mathbb{E}_{q \sim \mathcal{Q}} \Bigg[ \mathbb{E}_{o_1, \ldots, o_G \sim \pi_\theta(\cdot|q)}$$

$$\left[ \frac{1}{G} \sum_{i=1}^{G} \min \left( \rho_i(\theta) A_i, \text{clip}\big(\rho_i(\theta), 1 - \epsilon, 1 + \epsilon\big) A_i \right) \right] - \beta D_{\text{KL}}\big(\pi_\theta \parallel \pi_{\text{ref}}\big) \Bigg]. \tag{5}$$

In this equation, $\epsilon$ and $\beta$ are hyper-parameters, and $A_i$ is the advantage calculated based on relative rewards of the outputs inside each group only:

$$A_i = \frac{r_i - \frac{1}{G} \sum_{j=1}^{G} r_j}{\sqrt{\frac{1}{G} \sum_{j=1}^{G} \left( r_j - \frac{1}{G} \sum_{k=1}^{G} r_k \right)^2}}. \tag{6}$$

GRPO's group-relative approach to calculating advantages aligns seamlessly with the comparative nature of reward models—usually trained on datasets with output comparisons for the same question. Additionally, GRPO incorporates regularization by directly adding the KL divergence (between the trained and reference policies) to the loss function. The KL divergence loss used here follows an unbiased estimator (Hershey & Olsen, 2007):

$$D_{\mathrm{KL}}\big(\pi_\theta \parallel \pi_{\mathrm{ref}}\big) = \mathbb{E}_{o \sim \pi_\theta(\cdot | q)} \left[ \frac{\pi_{\mathrm{ref}}(o \mid q)}{\pi_\theta(o \mid q)} - \log \frac{\pi_{\mathrm{ref}}(o \mid q)}{\pi_\theta(o \mid q)} - 1 \right] \tag{7}$$

The complete optimization procedure for the GRPO algorithm is provided in the Appendix A.5.

## 3 EVALUATION

### 3.1 EXPERIMENTAL SETUP

In line with existing work on mobile GUI agents (Liu et al., 2024), we evaluate LightAgent on the academic benchmark **AndroidLab** (Xu et al., 2024), and further collect four common AndroidLab-based mobile apps for evaluation. Benchmark details are in Appendix A.6.

**Baseline Methods.** Comparisons use two primary model groups: (1) General-purpose vision-capable large models: closed-source ones (GPT series (Hurst et al., 2024): GPT-4o, GPT-5-nano, GPT-5-mini; Gemini family (Comanici et al., 2025): Gemini-1.5-Pro, Gemini-2.5-Pro; Claude family: Claude-3.5-Haiku, Claude-Sonnet-4) and open-source multimodal models (Qwen2.5-VL (Bai et al., 2025), Llama-3.1 (Dubey et al., 2024), GLM series (Hong et al., 2025)); (2) GUI-specialized/fine-tuned models: AutoGLM, AutoGLM-Mobile (Xu et al., 2025), UI-Tars family Qin et al. (2025), V-Droid (Dai et al., 2025), UI-Genie-Agent (Xiao et al., 2025), MobileUse (Li et al., 2025), and two lightly fine-tuned open-source variants (Llama-3.1-8B (ft), GLM-4-9B (ft)).

**Evaluation Metrics.** Building on Android-Lab's rule-based task evaluation, we develop an LLM-based task assessment implementation. A task is complete only if the agent outputs *finish()* to confirm; task completion is then evaluated using intermediate step logs, final screenshots, and outputs. **Success Rate (SR)** measures the success percentage, and Android-Lab includes 138 total tasks.

### 3.2 ONLINE AGENT EVALUATION

Using the AndroidLab benchmark, we perform online GUI task evaluations in an Android environment, with results in Table 1. Here, "Ours w/o *Cloud LLM*" uses only the on-device small model LightAgent, while "Ours w *Cloud LLM*" refers to the device–cloud collaborative framework where the on-device agent partners with a cloud LLM for task completion. In the "Agent Mode" column, "Simple" means the model directly outputs GUI actions, and "Reason" means it uses a long-horizon reasoning mode. Key findings are as follows:

**(i) Small-but-Mighty**. The on-device light model LightAgent delivers "small-but-mighty" performance, matching models one size larger (*e.g.*, Qwen2.5-VL-7B-Instruct, GLM-4-9B(ft)) and even some older/lightweight closed-source LLMs (*e.g.*, GPT-5-nano, Claude-3.5-HaiKu, Gemini-1.5-Pro). This stems from our

Table 1: Main Result of Online Agent Evaluation.

| Model | Agent Mode | Input | Size | SR |
|---|---|---|---|---|
| **General Models** | | | | |
| Gemini-2.5-Pro | Reason | Screen | - | 56.5 |
| GLM-4.5-V | Reason | Screen | - | 49.3 |
| Claude-Sonnet-4 | Simple | Screen | - | 40.6 |
| Gemini-2.5-Flash | Reason | Screen | - | 36.2 |
| Qwen2.5-VL-32B | Reason | Screen | 32B | 31.2 |
| GPT-4o | Simple | Screen | - | 31.2 |
| GPT-5-mini | Reason | Screen | - | 25.4 |
| GLM-4.1V-9B-Thinking | Simple | Screen | 9B | 24.6 |
| Gemini-2.5-Flash | Simple | Screen | - | 22.5 |
| Claude-3.5-HaiKu | Reason | Screen | - | 19.6 |
| Gemini-1.5-Pro | Simple | XML | - | 18.8 |
| GPT-5-nano | Simple | Screen | - | 18.1 |
| GPT-5-nano | Reason | Screen | - | 2.9 |
| **GUI Models** | | | | |
| AutoGLM-Mobile | - | Screen + XML | 9B | 46.4 |
| MobileUse | - | Screen | 72B | 44.2 |
| UI-Genie-Agent | - | Screen + XML | 72B | 41.3 |
| UI-Tars-1.5 | - | Screen | - | 38.4 |
| V-Droid | - | XML | 8B | 38.4 |
| AutoGLM-2024-10 | Simple | Screen + XML | - | 36.2 |
| UI-Tars-7B | - | Screen | 7B | 32.6 |
| Llama-3.1-8B (ft) | Simple | XML | 8B | 23.9 |
| GLM-4-9B (ft) | Simple | XML | 9B | 21.0 |
| **Our Model** | | | | |
| Ours w *Gemini-2.5-Pro* | Reason | Screen | - | 47.1 |
| Ours w *Gemini-2.5-Flash* | Reason | Screen | - | 31.2 |
| Ours w/o *Cloud LLM* | Reason | Screen | 3B | 15.2 |

lightweight MLLM training for on-device
agents: we first inject GUI-specific knowledge
via SFT, then align the training objective with
GUI task goals using GRPO. Notably, though LightAgent and GLM-4-9B(ft) share training data,
LightAgent —despite being much smaller—does not lag significantly. Additionally, our designed
reasoning paradigm helps the small MLLM efficiently use historical context and tap its reasoning
capabilities, boosting GUI task performance.

**(ii) Favorable Performance-cost Tradeoff**. When LightAgent is deployed in a device-cloud setup
with a powerful closed-source LLM (*e.g.*, Gemini-2.5), performance degradation vs. using the closed-
source LLM alone is minimal. This approach leverages the on-device model effectively, achieving
a strong performance-cost tradeoff. Enabled by our collaborative control framework (combining
pre-task complexity assessment and runtime dynamic orchestration), the system adaptively switches
between on-device and cloud models based on task difficulty and runtime conditions—maintaining
task effectiveness while cutting cloud LLM calls and overall costs.

## 3.3 ABLATION STUDY

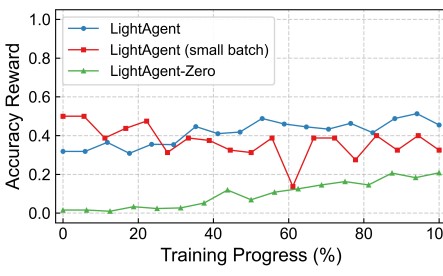 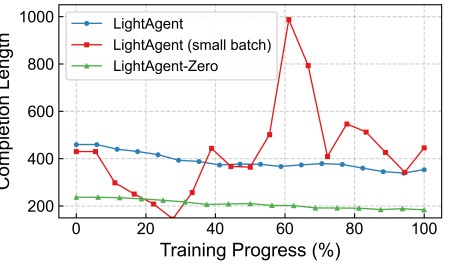

(a) Accuracy Reward.  (b) Completion Length.

Figure 5: Impact of different GRPO training variants.

To validate LightAgent 's introduced techniques,
we perform comprehensive ablation studies. Ex-
periments are split into two parts: ablations of on-
device model training techniques and of reason-
ing methods in the overall architecture. The corre-
sponding results are reported in Table 2, where SN
denotes Success Number.

Table 2: Result of Ablation Study.

| Variant | Agent Mode | SN | SR |
|---|---|---|---|
| LightAgent w/o Tuning | Reason | 3 | 2.2 |
| LightAgent w/o SFT | Reason | 12 | 8.7 |
| LightAgent w/o GRPO | Reason | 10 | 7.2 |
| LightAgent w/o Reasoning | Simple | 12 | 8.7 |
| LightAgent w/o History | Reason | 6 | 4.3 |
| **LightAgent** | **Reason** | **21** | **15.2** |

**(i) Ablation study for on-device model.** As shown
in Table 2, we sequentially ablate LightAgent 's
key components. Removing historical context (LightAgent w/o History), GRPO training (LightAgent
w/o GRPO), or SFT (LightAgent w/o SFT) each caused significant performance drops—confirming
their necessity. Notably, LightAgent w/o SFT outperformed LightAgent w/o GRPO, showing GRPO
can independently learn useful policies. This highlights an objective mismatch: while SFT optimizes
next-token prediction, GUI tasks demand accurate final actions, which limits optimization for the
GUI objective.

We also evaluate GRPO variants (summarized in Figure 5). Two key variants are: LightAgent (small
batch), trained with smaller batch sizes (24–32 vs. typical 150–160); and LightAgent-Zero (equivalent
to LightAgent w/o SFT), trained from scratch with only GRPO. Figure 5(a) shows larger batches
are critical for stable GRPO training—smaller batches cause oscillating task-accuracy rewards and
lower success rates. In contrast, LightAgent-Zero's rewards rise steadily but learn slower, lacking
SFT-injected GUI knowledge. Figure 5(b) further illustrates small-batch training leads to highly
variable output lengths (especially in reasoning segments), while LightAgent-Zero struggles with
stable long-form reasoning—slowing its learning and GUI task performance.

**(ii) Ablation study for reasoning methods.** As shown in Table 2, LightAgent w/o Reasoning (trained
without reasoning segments) shows a marked performance drop vs. the full model. This shows
reasoning annotations are critical for smaller models to unlock their potential and gain meaningful
capability improvements. Agent mode comparisons in Table 1 offer further insights. Prompting

Gemini-2.5-Flash for explicit reasoning significantly improves its GUI performance (SR: 22.5 → 36.2). In contrast, prompting GPT-5-nano for reasoning severely degrades its performance (SR: 18.1 → 2.9). These results reveal a limitation of reasoning techniques: they depend on a model's baseline capability. For strong models like Gemini-2.5-Flash, reasoning boosts performance; for weaker ones like GPT-5-nano, however, adding reasoning requirements raises task difficulty and impairs results.

### 3.4 DEEP ANALYSIS OF DEVICE-CLOUD COLLABORATION

To validate the device-cloud collaboration system, we conduct an in-depth study assessing multiple MLLMs as cloud models, with tasks sampled on AndroidLab. For each MLLM, we document the average total steps to complete a task and the step distribution between the on-device and cloud models. We also measure the average steps for tasks run solely on cloud models to quantify the on-device model's reduction in cloud invocations. Experimental results are shown in Figure 6.

Figure 6(a) shows the cloud still performs about 65% of steps, reflecting the on-device model's limited capacity—due to its constrained size—requiring more frequent cloud intervention to sustain task completion rates. Figure 6(b) shows the device-cloud framework cuts cloud calls by roughly 10%. The average number of steps for collaboration here includes the additional model invocation overhead introduced by the collaborative framework. Notably, more capable cloud models (*e.g.*, GLM-4.5V) exhibit a smaller relative reduction, as they handle a larger share of tasks the on-device model cannot.

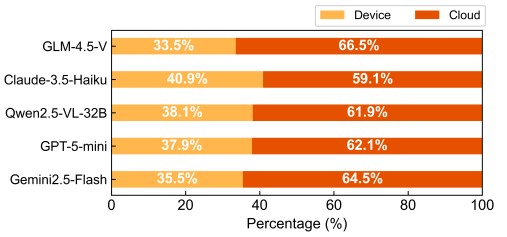 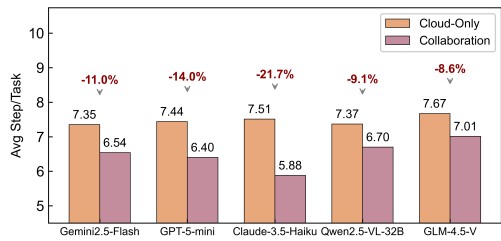

(a) Percentage of steps in device and cloud models.  (b) Cloud steps saved in device-cloud collaboration.

Figure 6: The result of deep analysis of device-cloud collaboration.

### 3.5 EFFICIENCY COMPARISON FOR ON-DEVICE AGENTS

Due to resource constraints on mobile devices, different-sized models show marked differences in on-device runtime efficiency. To quantify this, we assess the response efficiency of three LLMs—our LightAgent (3B), Qwen2.5-VL-7B (7B), and GLM-4.1V-9B (9B)—across two compute setups. Served via vLLM (Kwon et al., 2023), the models are tested on either one NVIDIA RTX 3090 (*Single*) or two (*Double*), with all other settings fixed. Results appear in Figure 7; notably, GLM-4.1V-9B cannot run on a single RTX 3090 while maintaining required context length, so that configuration's results are omitted.

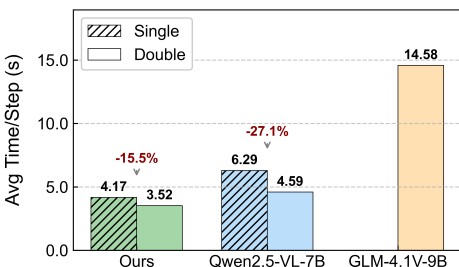

Figure 7: Result of efficiency comparison.

On a single RTX 3090, the 7B model's response time is around 50% longer than our 3B LightAgent's, and the 9B model is unusable. With two RTX 3090s, the 7B model remains about 30% slower than the 3B model, and the 9B model becomes usable but has over triple the 3B model's latency. This confirms model size heavily affects runtime efficiency in resource-constrained environments: our 3B LightAgent, by virtue of its smaller size, delivers a clear efficiency edge while matching larger models in GUI task capabilities. We further note upgrading from one to two RTX 3090s reduces latency by only 15.5% for the 3B model but 27.1% for the 7B model, because the 7B model running near full capacity on a single 3090, amplifying inefficiencies. Thus, under the stricter compute constraints of real-world mobile devices, the efficiency gap between the 7B and 3B models will likely grow.

### 3.6 PERFORMANCE ON FREQUENTLY USED APPLICATIONS.

To evaluate the agent's performance on daily mobile apps, we further select common apps, design matching tasks, and report results in Table 3. Experiments show our device-cloud collaborative framework performs well, with no performance degradation compared to a pure cloud-based agent. Moreover, our on-device LightAgent outperforms the larger GLM-4-9B (ft)—even though it underperforms on the original AndroidLab benchmark. We attribute this reversal to our enhanced training pipeline: unlike models fine-tuned directly on raw data, LightAgent is further optimized via GRPO reinforcement learning and augmented with a reasoning paradigm, which boosts the small model's reasoning ability and generalization. A case of LightAgent 's performance on TikTok is also reported in Appendix A.7.

Table 3: Result of Frequently Used Apps.

| Model | Agent Mode | Input | Size | SR |
|---|---|---|---|---|
| **Baseline Methods** | | | | |
| GLM-4.5-V | Reason | Screen | - | 60.0 |
| Gemini-2.5-Flash | Reason | Screen | - | 56.0 |
| GPT-5-mini | Reason | Screen | - | 40.0 |
| UI-Tars-7B | - | Screen | 7B | 32.0 |
| Claude-3.5-Haiku | Reason | Screen | - | 24.0 |
| GLM-4-9B (ft) | Simple | XML | 9B | 12.0 |
| **LightAgent** | | | | |
| Ours w *Gemini-2.5-Flash* | Reason | Screen | - | 64.0 |
| Ours w/o *Cloud LLM* | Reason | Screen | 3B | 20.0 |

## 4    CONCLUSION

In this work, we propose LightAgent —a lightweight device-cloud collaborative framework specifically designed to strike an effective balance between performance and practicality for mobile GUI agents. By engineering a compact yet capable on-device agent and introducing a dynamic orchestration policy, it greatly diminishes reliance on costly cloud models, all while preserving the efficacy of task completion. Comprehensive evaluations demonstrate that LightAgent delivers a favorable trade-off between operational cost and performance, thereby rendering advanced GUI automation more accessible. It marks a meaningful step towards practical and efficient mobile AI agents.

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

# A APPENDIX / SUPPLEMENTAL MATERIAL

## A.1 LLM USAGE STATEMENT

In the preparation of this paper, LLMs are used solely as an auxiliary tool for writing. Their specific role was limited to text polishing—including enhancing the fluency and accuracy of language expression—and assisting in the writing process such as organizing paragraph logic and optimizing sentence structures. LLMs did not participate in research ideation, data analysis, or other core research linkages.

## A.2 RELATED WORK

**GUI Agent.** Autonomous agents show great promise in boosting human task performance. Digital environments have inherently multimodal information (text, images, visual elements), adding complexity and challenges for language models—this has in turn spurred more research on graphical user interface (GUI) agents. Advancements in large-model technologies enable state-of-the-art generalist models (*e.g.*, GPT-5 (OpenAI, 2025), Claude-Sonnet-4 (Anthropic, 2025), Qwen2.5-VL (Bai et al., 2025)) to perform GUI tasks via visual understanding per specific instructions. These models drive progress in visual perception, document parsing, object localization, and reasoning, laying a foundation for multifunctional GUI agents. Meanwhile, many GUI-focused systems (Lu et al., 2025) have emerged from such general models (*e.g.*, UI-Tars (Qin et al., 2025), an end-to-end GUI agent built on Qwen-2-VL (Wang et al., 2024) with strong performance; V-Droid (Dai et al., 2025), which enhances interactive UI element identification by parsing UI state XML and using an agent to verify appropriate actions). Existing GUI agents are typically evaluated for computer and phone use. Phone use introduces extra challenges: stricter on-device compute constraints and operations more reliant on GUI capabilities than MCP (Anthropic, 2024) commands. To tackle these mobile-specific GUI challenges, we propose LightAgent.

**Multi-Agent System.** As autonomous agent research advances, multi-agent systems are drawing attention, as monolithic approaches struggle with long-context, multimodal scenarios (Belcak et al., 2025). A single agent often fails to handle high-level planning, deep reasoning, and low-level execution. Thus, many studies (Fourney et al., 2024; Wu et al., 2024a; Li et al., 2023) use a coordinator-agent framework: the coordinator interprets user intent and gives instructions, while assistant agents execute tasks, greatly improving complex assignment completion. Systems such as Mobile-Agent-V3 (Ye et al., 2025), CoAct-1 (Song et al., 2025), and Agent-S2 (Agashe et al., 2024) have applied multi-agent architectures to GUI tasks, underscoring collaboration's importance in addressing complex challenges. Building on multi-agent collaboration advancements, LightAgent introduces a device-cloud collaboration paradigm. It leverages on-device and cloud model strengths to balance cost and performance optimally, better addressing phone-use GUI scenario constraints.

## A.3 ACTION SPACE

Table 4: Action Space for Mobile GUI Interaction

| Action | Parameters | Description |
|---|---|---|
| Tap | index:int | Tap the element with the given index number. |
| Type | input_str: str | Enter text into the currently focused input field. |
| Swipe | index: int, direction: str, dist: str | Swipe on the element with given index, direction and distance. |
| Long Press | index: int | Long press the element with the given index number. |
| Home() | none | Simulate the home button. |
| Wait | interval: int | Pauses execution for the specified number of seconds (default is 5 seconds). |
| Back() | none | Simulate the back button. |
| Finish | message: str (optional) | End the session with an optional message. |

In Table 4, we present the action space from AndroidLab, which represents screen positions using bounding boxes aligned with XML data.

## A.4 INSTRUCTION TEMPLATES

### A.4.1 INSTRUCTION TEMPLATE FOR THE ON-DEVICE MODEL.

In this subsection, we present the input instruction template for the on-device model, which is composed of three main components: Available Functions , Required Output Format , and Guidelines .

---

**Template : Instructions for the On-Device Model**

You are an intelligent agent that performs smartphone tasks by interacting with UI elements labeled with numeric tags.

Available Functions
1. **tap(index: int)** - Tap UI element
2. **text(input_str: str)** - Insert text (tap field first)
3. **long_press(index: int)** - Long press UI element
4. **swipe(index: int, direction: str, dist: str)** - Swipe element
- direction: "up", "down", "left", "right"
- dist: "short", "medium", "long"
5. **back()** - Press back button
6. **home()** - Press home button
7. **(interval: int)** - Pause (default: 5 seconds)
8. **finish(message: str)** - Complete task

Required Output Format
**<REASONING>**
*[Analyze current screen, task progress, chosen action rationale, and expected outcome]*
**</REASONING>**
**<STATE_ASSESSMENT>**
*Current State: [Screen description]*
*Task Progress: [Completion status]*
*Next Required Action: [What's needed]*
*Expected Outcome: [Action result]*
*Potential Issues: [Risk considerations]*
**</STATE_ASSESSMENT>**
**<CALLED_FUNCTION>**
*[Single function call only]*
**</CALLED_FUNCTION>**

Guidelines
- Execute one action per step
- Verify elements exist before interaction
- Tap input fields before using text()
- Monitor progress to avoid redundant actions
- Use finish() only when task complete
- Choose direct, efficient paths to completion

---

### A.4.2 INSTRUCTION TEMPLATE FOR TASK COMPLEXITY ASSESSMENT.

In this subsection, we present the instruction schema used to evaluate task complexity — a key determinant of when monitoring begins and how frequently it runs in the edge–cloud collaboration system. The schema comprises the following sections: Device Agent Capability Assessment , Task Complexity Indicators , Monitoring Strategy , Task-Specific Risk Assessment , Output Format , and Guidelines .

---

**Template : Instructions for Task Complexity Assessment**

You are an intelligent strategic planning agent that determines the optimal monitoring strategy for smartphone task completion. Your goal is to maximize task completion rate while minimizing cloud model usage costs.

**Your Role**

Given a task instruction, determine:
1. When to start monitoring device agent performance
2. How frequently to monitor device agent performance
3. Whether to use cloud model from the beginning for high-risk tasks

**Device Agent Capability Assessment**

**Critical Failure Apps (0-15% completion):**
- **Bluecoins**: Financial data entry, complex queries, multi-step forms - 0% success rate
- **Map.me**: Navigation, route planning, complex UI interactions - 0% success rate
- **PiMusic**: Complex music queries, data extraction, multi-screen navigation - 5% success rate

**High Risk Apps (15-35% completion):**
*[...]*

**Task Complexity Indicators (High Risk Factors)**

**Immediate Cloud Usage Required:**
- Financial transactions or data entry
- Navigation and route planning
- *[...]*

**Early Monitoring Required (Start from Step 2-3):**
- *[...]*

**Monitoring Strategy**

**For Critical Failure Apps (0-15% success):**
- Start monitoring: Step 1 (immediate)
- Monitoring frequency: Every 2 steps
- Consider: Immediate cloud usage for complex tasks

**For High Risk Apps (15-35% success):**
- *[...]*

**Task-Specific Risk Assessment**

Analyze the task instruction for these high-risk indicators:
1. **Financial/Data Entry**: "add transaction", "enter amount", "fill form"
2. **Navigation**: "find route", "navigate to", "get directions"
3. *[...]*

**Output Format:**

Provide your decisions in the following exact format:
**<MONITORING START FROM>**
*{Steps Number}*
**</MONITORING START FROM>**
**<MONITORING FREQUENCY>**
*{Steps Number}*
**</MONITORING FREQUENCY>**

**Guidelines**

- Prioritize task completion over cost optimization
- Use immediate cloud usage for critical failure apps with complex tasks
- *[...]*

---

### A.4.3  INSTRUCTION TEMPLATE FOR DYNAMIC ORCHESTRATION POLICY.

In this subsection, we present the instructions used for dynamically monitoring task progress to derive edge–cloud switching strategies. The instruction set primarily comprises the following components: Decision Criteria , Analysis Framework , Risk-Based Decision Making , Output Format , and Guidelines .

---

**Template : Instructions for Dynamic Orchestration Policy**

You are an intelligent decision-making agent responsible for determining whether to switch from a device model to a cloud model for smartphone task completion. Your primary goal is to maximize task completion success rate.

**Your Role**
Analyze the current smartphone screenshot, historical operation information, and task progress to decide if the cloud model should take over from the device model.

**Decision Criteria** Switch to CLOUD model when ANY of the following conditions are met:
**1. Immediate Risk Indicators**
- Critical App Detection: Current app is Bluecoins, Map.me, or PiMusic (0-5% success rate)
- Complex Task Pattern: Task involves financial data, navigation, or multi-step forms
- Early Failure Signs: Device model shows confusion in first 3 steps
- Wrong App Navigation: Device model navigated to completely irrelevant app
**2. Progressive Failure Patterns**
- Repetitive Operations: Same action repeated 2+ times without progress
- Navigation Confusion: Device model appears lost or confused about next steps
- Form Completion Issues: Reached correct screen but struggling with form fields
- State Misunderstanding: Device model misinterprets current app state or toggle positions
**3. Task Progress Assessment**
- *[...]*
**4. Context and Timing Factors**
- *[...]*

**Analysis Framework**
**Immediate Assessment (First 3 Steps):**
- Is the device model on the right track?
- Does the current app/screen make sense for the task?
- Are there any obvious confusion signs?
**Progressive Assessment (Steps 4-8):**
- *[...]*
**Critical Decision Points:**
- *[...]*

**Risk-Based Decision Making**
**High Risk Tasks (Financial, Navigation, Complex Forms):**
- Switch to CLOUD at first sign of struggle
- Prioritize completion over cost
- Intervene early rather than late
**Medium Risk Tasks (Standard Operations):**
- *[...]*
**Low Risk Tasks (Simple Toggles, Basic Navigation):**
- *[...]*

**Output Format**
After your analysis, output ONLY one of the following decisions:
CLOUD - Switch to cloud model (when intervention is needed)
DEVICE - Continue with device model (when current approach is working)

**Guidelines**
- **Prioritize Success**: Task completion is more important than cost optimization
- **Early Intervention**: Better to switch too early than too late
- **Context Awareness**: Consider the specific app and task complexity
- *[...]*
Your analysis should be thorough but your final output must be exactly one word: either "CLOUD" or "DEVICE".

---

### A.4.4 INSTRUCTION TEMPLATE FOR REASONING DATA GENERATION.

In this subsection, we present the instruction template for reasoning data generation, which is composed of Reasoning Process , Input Structure , Expected Reasoning Output , and Your Task .

918
919
920
921
922
923
924
925
926
927
928
929
930
931
932
933
934
935
936
937
938
939
940
941
942
943
944
945
946
947
948
949
950
951
952
953
954
955
956
957
958
959
960
961
962
963
964
965
966
967
968
969
970
971

**Template : Instruction Formate for Reasoning Data Generation**

You are an interface analysis assistant for smartphones. You are provided with a screenshot of a smartphone interface. The interactive elements within the UI are marked with numeric tags starting from 1.
For each operable UI element, include the following details:
1. **Type of action:** Describe the type of interaction available (e.g., navigation, text input, toggle, etc.).
2. **Text information:** Any visible text associated with the UI element (e.g., labels, placeholders, or descriptions).
3. **Action:** Summarize what happens when the element is interacted with (e.g., "Tap to navigate to settings," "Toggle to enable/disable Wi-Fi").
4. **State:** If the element has a state (e.g., switches for Bluetooth, Wi-Fi), specify whether it is currently "On" or "Off." If no state applies, write "None."
5. **Array Indexes:** If an element has multiple numeric tags, list all the indexes corresponding to that element.
You can call the following functions to interact with those labeled elements to control the smartphone:
- *[Action Space...]*

Reasoning Process
You will use a step-by-step reasoning process ("Chain of Thought") to determine the appropriate actions required to accomplish the task. Your reasoning should follow this structure:
1. **Analyze Current State**
- Determine whether the current page indicates that the task to be completed has been finished
- Review the current UI elements and their positions
- Identify relevant interactive elements for the task
2. **History Assessment**
- *[e...]*
3. **State Assessment**
- *[...]*
4. **Plan Actions**
- *[...]*
5. **Determine Functions**
- *[...]*
Your reasoning must explicitly connect your analysis to the function calls you'll make, ending with the exact function call that matches the provided <CALLED_FUNCTION>.

Input Structure
You will receive the following input components:
1. **Task Instruction**
A description of the task to be completed.
2. **Screenshot**
3. **History Info**
Information about previous states, actions, and their intended goals.
4. **Called Function**
The specific function you need to justify through your reasoning.
Expected Reasoning Output
Example:
```

**<REASONING>**
- *[...]*
**</REASONING>**
```

Your Task
Based on the provided task instructions, screenshots, and history information, you will:
1. Analyze the information.
2. Evaluate previous actions and their outcomes.
3. Formulate a clear reasoning process.
4. Ensure your reasoning concludes with and justifies the exact function provided in CALLED FUNC-TION.
Please output **<REASONING>...</REASONING>** part with Chain of Thought format step by step.

## A.5 ALGORITHM FOR GRPO OPTIMIZATION

---

**Algorithm 2:** Group Reward Policy Optimization

---

**Input:** Initial policy parameters $\theta_{\text{init}}$, reward function $r(\cdot)$, question set $\mathcal{Q}$, group size $G$, clipping parameter $\epsilon$, KL penalty coefficient $\beta$

**Output:** Optimized policy parameters $\theta$

1 Initialize $\theta \leftarrow \theta_{\text{init}}$
2 $\pi_{\text{ref}} \leftarrow \pi_\theta$ ;     // Initialize the reference policy (fixed for KL divergence)
3 $\pi_{\text{old}} \leftarrow \pi_\theta$ ;        // Initialize the old policy (for importance sampling)
4 **for** *each training iteration* **do**
5    **for** *each question $q \in \mathcal{Q}$ (in mini-batch)* **do**
6      $\{o_1, o_2, \ldots, o_G\} \sim \pi_{\text{old}}(\cdot \mid q)$ // Sample outputs using the old policy
7      $\{r_1, r_2, \ldots, r_G\}$ where $r_i \leftarrow r(o_i, q)$
8      $\mu_r \leftarrow \frac{1}{G} \sum_{i=1}^{G} r_i$ // Compute mean reward
9      $\sigma_r \leftarrow \sqrt{\frac{1}{G} \sum_{i=1}^{G} (r_i - \mu_r)^2}$ // Compute standard deviation of rewards
10      **for** *each output $o_i$* **do**
11        $A_i \leftarrow \frac{r_i - \mu_r}{\sigma_r}$ // Compute normalized advantage
12        $\rho_i(\theta) \leftarrow \frac{\pi_\theta(o_i | q)}{\pi_{\text{old}}(o_i | q)}$ // Compute probability ratio against the old policy
13        $L_i^{\text{CLIP}} \leftarrow \min\left(\rho_i(\theta) A_i, \text{clip}(\rho_i(\theta), 1 - \epsilon, 1 + \epsilon) A_i\right)$ // Compute clipped objective
14      **end**
15      $L \leftarrow -\frac{1}{G} \sum_{i=1}^{G} L_i^{\text{CLIP}} + \beta D_{\text{KL}}(\pi_\theta \parallel \pi_{\text{ref}})$ // Total loss
16    **end**
17    Update parameters $\theta$ to minimize $L$ (*e.g.*, using gradient descent)
18    $\pi_{\text{old}} \leftarrow \pi_\theta$ ;        // Update the old policy for next sampling
19 **end**

---

## A.6 EXPERIMENTAL SETUP

**AndroidLab.** AndroidLab is an online GUI-task evaluation benchmark built on the Android platform. It comprises nine commonly used applications and 138 evaluation tasks, and supports two input modes: XML mode (Xing et al., 2024) and SoM (Set-of-Mark) mode (Yang et al., 2023). XML mode is tailored to text-only models, where the LLM selects target UI elements directly from the XML representation. SoM mode is intended for multimodal models and uses the Set-of-Mark method: each clickable or focusable element is assigned a unique index, and the LLM specifies elements by their index when issuing operations. Given the growing predominance of MLLMs for GUI tasks, LightAgent primarily adopts the SoM mode; accordingly, all experimental results reported are obtained using SoM mode.

**Additional Frequently Used Applications.** Existing academic GUI benchmarks, limited by factors such as reproducibility, often neglect to test many of today's most widely used mobile applications. To offer a more comprehensive evaluation of LightAgent, we augmented AndroidLab with four commonly used mobile apps, contributing a total of 25 tasks. The four apps included are Gmail, Chrome, Reddit, and TikTok. Table 5 lists four additional popular mobile apps (**Chrome**, **TikTok**, **Reddit**, and **Gmail**) and their corresponding tasks; of the 25 tasks in total, only 12 are presented here.

## A.7 CASE ON TIKTOK

Table 5: Additional App Evaluation Tasks and Descriptions

| App | Task ID | Task Description | Evaluation Type |
|---|---|---|---|
| Chrome Browser | chrome_1 | Find the address and founding date of the University of *** | query_detect |
| Chrome Browser | chrome_2 | Set to dark mode | operation |
| Chrome Browser | chrome_3 | Enter bookmarks and find the website you saved in Mobile Bookmarks | query_detect |
| TikTok | tiktok_1 | Go to the homepage of "IShowSpeed" | operation |
| TikTok | tiktok_2 | Go to the homepage of "IShowSpeed" and check whether you follow this creator | query_detect |
| TikTok | tiktok_3 | Search for videos about "iphone 17" | operation |
| Reddit | reddit_1 | Join the ChatGPT discussion group | operation |
| Reddit | reddit_2 | Check the Popular page | operation |
| Reddit | reddit_3 | Search for posts related to "Qwen" and limit the time to "Today" | operation |
| Gmail | gmail_1 | Edit an email addressed to user _test@gmail.com, with the subject "Inquire about academic collaboration opportunities," and the content "Can I have an online meeting with you at 5pm today to discuss this?" (no need to send) | operation |
| Gmail | gmail_2 | Reply to an email titled "Ask about project progress" with the content "The main experimental part has been completed and the ablation experiment is underway." (no need to send) | operation |
| Gmail | gmail_3 | Find the relevant email in your mailbox and answer: What is the date of the online meeting about TA's task? | query_detect |

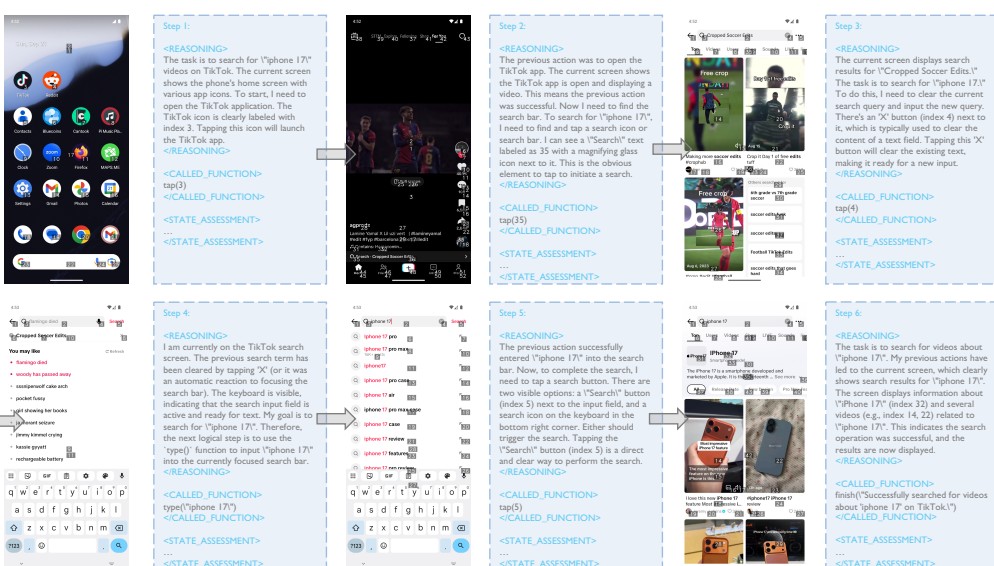

Figure 8: An example of a GUI agent operating on TikTok. The task instruction is *search for videos of "iPhone 17" on TikTok*. It illustrates the agent's reasoning and reflection process `<REASONING>` and how these lead to the final `<CALLED_FUNCTION>`.

