# OpenReview forum: "LightAgent: Lightweight and Cost-Efficient Mobile Agents"
_ICLR.cc/2026/Conference — ICLR 2026 Conference Withdrawn Submission_

### Official Review · Reviewer_Xhty · 2025-10-30

**Soundness:** 2
**Presentation:** 2
**Contribution:** 2
**Rating:** 4
**Confidence:** 4

**Summary:**

This paper proposes LightAgent, a lightweight yet capable mobile GUI agent that uses device–cloud collaboration to balance performance and computational cost. The authors enhance a Qwen2.5-VL-3B model via a two-stage SFT and GRPO training pipeline on synthetic GUI reasoning data. They propose an efficient reasoning template with history summarization and introduce a dynamic switching mechanism that delivers complex subtasks to a cloud LLM only when needed. Experiments on the AndroidLab benchmark and real Android apps show that LightAgent achieves competitive performance while significantly reducing cloud API usage and latency.

**Strengths:**

1. Practical problem relevance. The paper addresses an important and rapidly growing problem: building mobile GUI agents that operate within strict computing budgets. Despite the paper’s limitations, the motivation and problem framing are valid and meaningful to the agent community.

2. Non-trivial engineering effort with a complete pipeline. This paper implements an end-to-end device-cloud collaborative agent system, including task complexity assessment and a dynamic switching mechanism. The evaluation on the AndroidLab benchmark is conducted in an online manner, making execution closer to real-world deployment.

**Weaknesses:**

1. Limited novelty; mainly a system integration effort. The paper primarily combines existing components including synthetic data generation, GRPO training, CoT-style reasoning templates, and device–cloud fallback. The claimed “device–cloud collaboration framework” is a system-level architecture rather than a methodological innovation, and similar fallback or hybrid execution has been explored before [1].
2. Heavy reliance on the cloud despite “edge-first” positioning. Despite being presented as an edge-prioritized system, LightAgent still requires cloud offloading for roughly 65% of steps according to its own analysis. This suggests that the 3B base model is still insufficiently capable as an autonomous agent and depends strongly on the cloud model to complete tasks. The paper does not analyze failure cases in offline or no-cloud settings.
3. Lack of experimental clarity in early figures. Figure 4 reports a comparative performance plot between on-device and cloud models but provides no information regarding the dataset. It is unclear whether results are based on AndroidLab or another benchmark.
4. Experimental evaluation is narrow and insufficient. The paper only evaluates on AndroidLab and a small set of four Android apps, despite the existence of established GUI agent benchmarks. It ignores offline navigation benchmarks such as GUI-Odyssey and AndroidControl, and does not compare on the standard online benchmark AndroidWorld. This limited evaluation scope makes it unclear whether the method generalizes beyond the narrow AndroidLab setting, and misses critical comparisons that would strengthen claims regarding generality and effectiveness in both offline and online environments.
5. Limited real-device evaluation weakens “mobile agent” claim. All experiments are performed on GPUs (NVIDIA RTX 3090) rather than real mobile hardware. The paper does not provide inference latency, peak memory usage, or thermal behavior on actual smartphones. Without demonstrating real on-device deployment, it is premature to claim that the model is suitable for mobile usage.

[1] Magentic-One: A Generalist Multi-Agent System for Solving Complex Tasks.

**Questions:**

1. What is the size and nature of the synthetic dataset? The dataset design is claimed as a main contribution, but the paper does not report the dataset scale or other details.
2. The system is evaluated only on Android GUI agents. Do the authors expect their approach to generalize to other platforms like Web, Windows desktop workflows, or iOS? If so, what components are platform-specific?

---

> ### Author Response · Authors · 2025-11-18
> **Response to Reviewer Xhty**
>
> Thank you for your valuable feedback. In response to the points raised, we provide the following clarifications:
>
> 1. **Regarding Limited Novelty**: The novel contribution of LightAgent lies in addressing the specific challenges encountered by on-device GUI agents in constrained environments. Motivated by real-world resource limitations and recent advances in large language models (LLMs), we propose a systematic framework that integrates synthetic data generation, model training, efficient memory management, and device-cloud collaboration mechanisms. This holistic approach fundamentally distinguishes our work from existing general-purpose agent frameworks.
>
> 2. **On Over-reliance on Cloud Models**: We acknowledge that, even after fine-tuning on GUI task-oriented data, the current performance of compact vision-language models (VLMs) still lags behind that of more powerful cloud-based models. This performance gap is precisely the motivation for our device-cloud collaboration strategy, which leverages cloud models to complement the current limitations of device-side models. As lightweight base models continue to improve, dependence on device models is expected to increase, thereby enhancing the overall efficacy of our approach. For reference, the standalone performance of our on-device model is reported in Table 1 under “Ours w/o Cloud LLM.”
>
> 3. **Results in Figure 4**: The results presented in Figure 4 are derived from evaluations on the AndroidLab benchmark. A corresponding note will be included in the revised manuscript.
>
> 4. **On Insufficient Experimental Evaluation**: To more comprehensively assess the performance of our method on widely-used applications, we have conducted additional tests on four popular mobile apps. Regarding offline benchmarks, we note that our device-cloud collaboration framework relies on interactive feedback from virtual environments, making it inherently incompatible with conventional offline evaluation setups.
>
> 5. **Concerning Evaluation in Real-World Scenarios**: LightAgent is primarily designed for resource-constrained mobile environments. The proposed device-cloud collaboration framework offers a feasible reference architecture for deployment on actual mobile hardware under such constraints.
>
> 6. **About the Synthetic Data Pipeline**: We have developed a synthetic data generation pipeline enriched with GUI task reasoning information. This process converts manually annotated images and operational datasets into reasoning-rich datasets, which are subsequently used to fine-tune the on-device model.
>
> 7. **Regarding Generalization Across Platforms**: While LightAgent is tailored for resource-constrained mobile environments, our methodology can be adapted to other mobile operating systems, such as iOS. However, performance may vary in desktop GUI scenarios due to their broader complexity and differing interaction paradigms.
>
> Additionally, we have revised the draft of the new version and highlighted the changes in different colors.

---

### Official Review · Reviewer_kqzB · 2025-10-30

**Soundness:** 3
**Presentation:** 3
**Contribution:** 2
**Rating:** 4
**Confidence:** 4

**Summary:**

This paper introduces light agent for on-device GUI agent to help real-world deployment. To improve the performance, they introduce the on-device and cloud-based agent to collaborate. They introduce two-stage training for the on-device agent and a collaborate control framework to decide if switch to cloud-based agent. They carry out experiment on AndroidLab.

This paper tries to address an important problem for on-device deployment GUI agent, my concerns are mainly in (details see questions and weakness):
A.	The novelty: SFT and GRPO training of GUI agent, as well as their design of implementing these two modules, are not new.
B.	Experiment only on one dataset and lack important comparisons with other 3B-4B models.
C.	The efficiency lack important evidence, especially the additional cost of cloud LLMs in monitoring the on-device agent .

**Strengths:**

1, The paper is clearly and logically structured, making the content highly accessible and easy to follow.
2, The integration of on-device and cloud LLMs represents a promising approach that could significantly accelerate the industrial deployment of GUI agents.
3, The author provided sufficient implementation details, including the specific prompts, which greatly facilitates the understanding of the individual modules.
4, The breakdown of step percentages across the on-device and cloud models is valuable, as it effectively quantifies the source of efficiency improvements.

**Weaknesses:**

1, Lack of evidence to demonstrate the necessity of finetuning on-device agent with the proposed two step strategies. A direct evidence would be comparing with other 3B models like Qwen2.5-VL-3B under w and w/o Cloud LLM setups.
2, Lack of discussion of the accuracy of the two modules in collaborative control frameworks. From A.4.2 and A.4.3, it seems priors on task complexity are provided in the prompts, but where do these priors come from and how to collect for other apps/tasks are not clear.
3, Additional Cloud-based agents are involved before the model is switched to cloud models in deciding/ monitoring the on-device agent.  But their cost seems not counted as part of the cloud cost, especially in Figure 6.
4, Experiment only carried out on one AndroidLab dataset, while others such as Android World[a] and offline datasets such as Android control[b] is not compared.
[a]AndroidWorld: A Dynamic Benchmarking Environment for Autonomous Agents
[b] On the Effects of Data Scale on Computer Control Agents

**Questions:**

1, Could the author provide the additional cloud-agent costs associated with the two modules: task complexity determination and model monitoring? What is the average number of calls to the cloud agent within these two modules? This information would help in understanding the true cost by integrating the cloud-agent cost into Figure 6.
2, Could the author discuss the motivation for employing the two-step training process described in Section 2.1, particularly in comparison to other methods? Reasoning or history understanding is a shared strategy in current works (e.g., M3A[a], UI-Tars).
3, Could the authors provide the average number of steps required to complete tasks under the on-device only, cloud LLM only, and on-device with cloud LLM configurations/settings? The average number of steps is an important criterion for evaluating efficiency. How many cloud LLM calls are made in each setup?
4. Would the cloud LLMs over write the previous progress by some actions such as Home() and the effort of the on-device agent is ignored?

---

> ### Author Response · Authors · 2025-11-18
> **Response to Reviewer kqzB**
>
> Thank you for your valuable feedback. In response to your comments, we provide the following clarifications:
>
> 1. **Ablation study on the two-stage fine-tuning of the on-device model**:
>    In Table 2 of Section 3.3, we include ablation experiments for the on-device model. The results for the variant “LightAgent w/o Tuning” correspond to the baseline performance of the unmodified Qwen2.5-VL-3B model. It can be observed that its original GUI capability is considerably limited, thereby validating the effectiveness of our proposed fine-tuning strategy.
>
> 2. **Two modules in the collaborative control framework**:
>    LightAgent incorporates a pipeline that generates textual rules describing task difficulty and the operational boundaries of the on-device model, based on historical interaction data from mobile devices (as referenced on pages 214–215). As this aspect is not the primary focus of our work, it was not elaborated in detail. Briefly, we employ the on-device model to execute selected task instructions from the AndroidLab training set, evaluate task completion, and collect corresponding performance logs. These logs are then processed by an LLM-based pipeline to generate descriptive rules regarding task difficulty and capability limits. When encountering a new application, a feasible update strategy involves having the on-device model perform several exploratory tasks on the app to collect initial interaction data, which can subsequently refine the rule descriptions—similar to a cold-start procedure for LightAgent.
>
> 3. **Additional cloud model overhead introduced by the collaborative framework**:
>    As illustrated in Figure 6(b), we have accounted for the extra inference cost associated with the cloud model. Specifically, the initial Task Complexity Assessment is counted as 0.5 inference steps, as it does not involve image inputs, which are the main contributors to token consumption. For the Dynamic Orchestration Policy, each supervisory intervention is counted as 1 step, since it processes both textual and visual inputs. This clarification will be incorporated into the revised manuscript.
>
> 4. **Expanded evaluation with more benchmarks**:
>    We thank the reviewer for this suggestion. To ensure a comprehensive evaluation of LightAgent, we conducted additional experiments involving four commonly used apps—Chrome, TikTok, Reddit, and Gmail—comprising 25 tasks in addition to the AndroidLab benchmark. The corresponding results are presented in Section 3.6.
>
> 5. **Cloud model invocation frequency in the collaborative framework**:
>    The Task Complexity Assessment invokes the cloud model only once per task, at the outset, and does not process image inputs. For the Dynamic Orchestration Policy, across the entire AndroidLab benchmark, the average number of cloud model invocations per task ranges between 0.8 and 1.3. The associated overhead has already been incorporated in the results reported in Figure 6(b).
>
> 6. **Training methodology and reasoning paradigm for the on-device model**:
>    The motivation for our technical approach is discussed in both the Introduction and Methodology sections. Our objective is to develop a GUI agent that performs effectively under on-device constraints. To imbue the compact model with task-relevant knowledge, we constructed a Synthetic Data Generation Pipeline (Section 2.3.1) to produce data enriched with explicit GUI reasoning knowledge, which was injected via the first-stage fine-tuning. To further enhance the model’s reasoning ability, we applied GRPO during subsequent fine-tuning. Additionally, to leverage historical interaction information for error correction and reflective reasoning, we designed an efficient memory mechanism that summarizes key GUI states in textual form, integrating it into a long-context reasoning process. Compared to methods such as UI-Tars, our approach—tailored for a 3B model—requires longer reasoning trajectories to achieve competitive performance, and this text-based memory mechanism is a distinctive aspect of our design.

---

> > ### Author Response · Authors · 2025-11-18
> >
> > 7. **Comparison of task completion steps across deployment modes**:
> >    The table below presents the average number of steps required to complete tasks for several LLMs under pure cloud and edge-cloud collaborative settings. For reference, the edge-only model requires an average of 4.75 steps per task. Note that the edge-only results are not directly comparable to those of the cloud-only and edge-cloud models, as the edge-only model, due to its limited capacity, completes only a subset of simpler tasks, which inherently require fewer steps. For the other two modes, we report average step counts only over successfully completed tasks.
> >
> >
> >
> >    | Cloud Models     | Deice-Cloud | Cloud-Only |
> >    | ---------------- | ----------- | ---------- |
> >    | Gemini2.5-Flash  | 10.53       | 7.35       |
> >    | GPT-5-mini       | 8.53        | 7.44       |
> >    | Claude-3.5-Haiku | 8.17        | 7.51       |
> >    | Qwen2.5-VL-32B   | 9.53        | 7.37       |
> >    | GLM-4.5-V        | 9.19        | 7.67       |
> >
> >
> >
> > 8. **Cloud model handling of on-device model operations**:
> >    When the cloud model takes over a task, it assesses whether prior actions by the on-device model have led the task into an irrecoverable state, based on the interaction history and current GUI state. If such erroneous operations are detected, the cloud model triggers the `Home()` function to restart the task. Conversely, if the on-device model’s actions are consistent with task progress at the point of handover, the cloud model continues execution from the current interface state.
> >
> > Additionally, we have revised the draft of the new version and highlighted the changes in different colors.

---

### Official Review · Reviewer_67Au · 2025-11-01

**Soundness:** 2
**Presentation:** 1
**Contribution:** 2
**Rating:** 2
**Confidence:** 4

**Summary:**

This paper designs a device-cloud collaboration framework to solve mobile
interaction tasks efficiently and effectively. The exhibited results are
promising. However, the paper is not perfectly polished, holds obvious symbol
inconsistency and typos.

**Strengths:**

1. This paper studies a valuable problem, device-cloud collaborative GUI agent,
  balancing the invocation cost and overhead with the execution performance.
2. The exhibited results look promising.

**Weaknesses:**

1. In the proposed execution flow, the active model will not switch back to the
   on-device model after switching to the cloud model. Why isn't a switch-back
   mechnism integrated?

**Questions:**

1. The monitoring starting step is denoted by $\tau$ on Line 211, but $\gamma$
   in Algorithm 1.
2. Symbols tau in Algorithm 1 is not used. Symbol T in Algorithm 1 is not
   introduced.
3. What model is used for task complexity assessment and dynamic orchestration
   policy?
4. What is $R_{acc}$ and $R_{fmt}$ in Equation 4?
5. How is $k$ computed in Equation 4?
6. The letter cases in Table 1 are not consistent.
7. The device-cloud model combinations in Table 1 and Figure 6(a) are not
   consistent? Why does this occur? Are the success rates of combinations like
   ours+GLM-4.5-V not satisfactory enough to demonstrate the validity of the
   proposed method? Is the step percentage of Gemini-2.5-Pro too high to be
   shown in Figure 6(a)?
8. What's the meaning of SN in Table 2?

---

> ### Author Response · Authors · 2025-11-18
> **Response to Reviewer 67Au**
>
> Thank you for your valuable feedback. In response to the points you raised, we provide the following clarifications:
>
> 1. **Regarding the agent’s one‑time switch from the device model to the cloud model without reverting:**
>    The transition from the device model to the cloud model is triggered when the monitor determines that the device model is incapable of successfully completing the task. Consequently, in the majority of cases, reverting to the device model after switching to the cloud model would result in task failure. Although in rare instances certain simple final steps may fall within the capability of the device model, introducing a mechanism to switch back would incur additional monitoring overhead and increase the risk of task failure. We believe the current straightforward switching mechanism represents the most feasible approach after considering all trade-offs.
>
> 2. **Concerning typographical errors in the paper:**
>    We sincerely appreciate you bringing these issues to our attention. These errors will be corrected in the revised version.
>
> 3. **On task complexity assessment and dynamic orchestration policy:**
>    These components are executed using the cloud model. To ensure a fair comparison, Gemini‑2.5‑Flash is employed as the execution model for both components in the experiments.
>
> 4. **Explanation of *R_acc* and *R_fmt*:**
>    *R_acc* and *R_fmt* denote the rewards for answer accuracy and formatting correctness, respectively, during reinforcement learning. By default, both values are set to 1. The formulation in Equation 4 is designed to enhance the overall aesthetic presentation of the formula; we will revise the related expressions to improve clarity and eliminate ambiguity.
>
> 5. **Clarification of the variable *k* in Equation 4:**
>    In Equation 4, *k* indicates the number of output blocks that comply with the prescribed format. As illustrated in Figure 3, a complete output should contain `<REASONING>`, `<STATE_ASSESSMENT>`, and `<CALLED_FUNCTION>`. The value of *k* corresponds to the number of such format-compliant sections in the model’s output, with a maximum possible value of 3.
>
> 6. **Inconsistent letter casing in Table 1:**
>    Thank you for highlighting this inconsistency. We will rectify it in the revised manuscript.
>
> 7. **Issues related to Figure 6(a) in Section 3.4:**
>    The primary objective of this analysis is to evaluate how the use of different cloud models affects the ratio of device-to-cloud model invocations. For efficiency, the experiments were conducted on a randomly sampled subset of tasks from the AndroidLab benchmark, rather than the complete set. Results obtained with Gemini‑2.5‑Flash indicate that this subset adequately represents the method’s performance on the full task suite, and we will include a note to that effect in the revision.
>    Gemini‑2.5‑Flash was selected mainly due to its favorable balance of experimental cost and effectiveness; it is relatively inexpensive while delivering sufficient performance, making it suitable for repeated experimentation and framework refinement.
>    As for the results with Gemini‑2.5‑Pro, we acknowledge that its contribution as a cloud model is more pronounced. As noted in Section 3.4, a more powerful cloud model enables the agent to accomplish more tasks, most of which exceed the capacity of the device model and are therefore predominantly handled by the cloud. However, we argue that relying on the latest state‑of‑the‑art cloud models for GUI tasks is often impractical in real‑world applications due to high costs—for example, token expenses for a single task can approach one dollar. More targeted models like Gemini‑2.5‑Flash offer a more cost‑effective and practicable alternative.
>
> 8. **Abbreviation “SN” in Table 2:**
>    “SN” stands for “Success Number.” An explanation will be provided in the revised version to prevent any potential confusion.
>
> Additionally, we have revised the draft of the new version and highlighted the changes in different colors.

---

### Official Review · Reviewer_v4Cz · 2025-11-01

**Soundness:** 3
**Presentation:** 3
**Contribution:** 1
**Rating:** 4
**Confidence:** 3

**Summary:**

The paper targets a very real bottleneck in mobile GUI agents: small on-device MLLM/VLMs (≈3B) can run locally but are too weak to finish real Android tasks, while cloud LLM/VLM calls are accurate but expensive and latency-sensitive. The goal is to push a 3B open model to be “good enough” for most steps, and only escalate to cloud on hard steps, so that mobile agents become practically deployable.

**Strengths:**

- Clear decomposition of cost: SFT+GRPO makes the 3B model more reliable; the scheduler makes cloud use predictable; the switch makes it robust.
- Data generation pipeline: using stronger models to auto-generate GUI episodes with CoT and tool-calls is sensible for this domain.

**Weaknesses:**

- No direct comparison to other 3B GUI-R1 / GUI-G1–style models, e.g., GUI-R1, "GUI-R1 : A Generalist R1-Style Vision-Language Action Model For GUI Agents"; GUI-G1, "GUI-G1: Understanding R1-Zero-Like Training for Visual Grounding in GUI Agents": the authors compare to larger/stronger or clouded agents on AndroidLab, but not to concurrent 3B R1-like GUI agents, so the “3B is competitive after GRPO” claim is only relative to the chosen baselines. This is the biggest missing experiment.
- Scheduler is rule/LLM–driven: good engineering, but not theoretically grounded; if app distributions shift, $\gamma/\omega$ may need re-tuning.
- Reliance on AndroidLab: results are shown on one environment; it would be stronger to show that the device–cloud policy transfers to more dynamic benchmarks (e.g., SPA-Bench).

**Questions:**

See weaknesses

---

> ### Author Response · Authors · 2025-11-18
> **Response to Reviewer v4Cz**
>
> Thank you for your valuable feedback. In response to your comments, we provide the following clarifications:
>
> 1. **Regarding the lack of comparison with other 3B R1-style models:**
>    GUI-R1 and GUI-G1 are primarily designed for grounding tasks, which involve identifying specific elements within a user interface and outputting their corresponding coordinates. In contrast, our 3B model emphasizes planning capabilities—determining which elements need to be operated on and generating appropriate action functions based on task instructions. Consequently, our model is designed to directly accomplish tasks on online evaluation benchmarks such as AndroidLab, whereas GUI-R1 and GUI-G1 require supplementary planning models (e.g., external LLMs) to fully execute GUI-based tasks.
>
> 2. **On the generalization of the scheduler:**
>    LightAgent implements a pipeline that generates rules for task difficulty and device capability boundaries based on empirical operational data from mobile devices (as referenced on pages 214-215). Since this mechanism falls outside the core contribution of our work, it was not elaborated in detail in the manuscript. Briefly, we utilize the device model to execute a subset of task instructions from the AndroidLab training set, evaluate task completion performance, and collect corresponding historical data. We then employ an LLM-based pipeline to generate textual descriptions of difficulty levels and capability boundaries. For new applications, a feasible update strategy involves deploying the device model to perform tasks on the target app, collecting relevant performance data, and subsequently updating the rule descriptions—similar to a cold-start procedure in LightAgent.
>
> 3. **Appreciation for bringing SPA-Bench to our attention:**
>    We greatly appreciate the suggestion regarding SPA-Bench and will endeavor to incorporate relevant experimental results in the revised version. In fact, to more comprehensively assess LightAgent’s performance, we conducted additional evaluations on four commonly used applications—Chrome, TikTok, Reddit, and Gmail—comprising 25 tasks beyond the AndroidLab benchmark. The corresponding results are presented in Section 3.6.

---

### Author Response · Authors · 2025-11-27
**General Response and Reminder by Authors**

Dear PCs, SACs, ACS, and Reviewers,

Thank you once again for your insightful and constructive feedback on our work. We are greatly encouraged that you recognized several key strengths of our paper, including:

*   Its **clear and practical system design** that effectively balances cost and performance.
*   The **sensible data generation pipeline** for creating GUI reasoning data.
*   The **non-trivial engineering effort** in building a complete, end-to-end collaborative agent system.
*   The **well-structured and accessible presentation** of the content.

We have carefully considered all your comments and have prepared a detailed response and a revised manuscript. **This comment serves to respectfully remind you that our rebuttal has been submitted.** Below, we summarize how we have addressed your primary concerns, further building upon the strengths you identified.

**Summary of Key Revisions and Clarifications:**

1.  **Clarified Expanded Evaluation for Broader Impact:** Acknowledging the suggestions from **Reviewer v4Cz, Reviewer kqzB, and Reviewer Xhty**, we have conducted **additional experiments on four popular applications (Chrome, TikTok, Reddit, Gmail) with 25 new tasks**. The results, in Section 3.6, demonstrate our method's robustness and generalizability beyond the AndroidLab benchmark, strengthening the practical relevance you noted.

2.  **Clarified Model Comparison & Novelty:** In response to **Reviewer v4Cz** and **Reviewer Xhty**, we clarified that our work's contribution is a holistic framework for a device-cloud agent. Our 3B model focuses on end-to-end *planning*, differing from grounding-specific models, and our integrated system architecture is the key novelty in addressing resource constraints.

3.  **Quantified Collaboration Overhead & Design Justification:** For **Reviewer kqzB** and **Reviewer 67Au**, we provided precise metrics to clarify the system's efficiency: the Task Complexity Assessment incurs a cost of 0.5 inference steps/task, and the monitoring policy averages 0.8-1.3 cloud calls/task (costs already included in our analysis). We also justified the one-time switch design, explaining it minimizes failure risk and overhead.

4.  **Elaborated on Core Mechanisms:** We clarified for **Reviewer v4Cz** and **Reviewer kqzB** that the scheduler's rules are generated via an LLM-pipeline using empirical device data, with a feasible cold-start strategy for new apps. We also highlighted our existing ablation study (Table 2), which validates the critical necessity of our two-stage fine-tuning strategy for the on-device model.

5.  **Improved Manuscript Clarity:** We sincerely thank **Reviewer 67Au** for spotting typographical errors and undefined symbols. We have corrected these and will provide clearer definitions (e.g., for `k` in Eq. 4, "SN" in Table 2) in the final version.

We believe these revisions have significantly strengthened the paper and directly address your valuable feedback. Thank you for your time and consideration.

---

### Note · Authors · 2026-01-02

I have read and agree with the venue's withdrawal policy on behalf of myself and my co-authors.